# The Pupa Stage Is the Most Sensitive to Hypoxia in *Drosophila melanogaster*

**DOI:** 10.3390/ijms25020710

**Published:** 2024-01-05

**Authors:** Tsering Stobdan, Nicholas J. Wen, Ying Lu-Bo, Dan Zhou, Gabriel G. Haddad

**Affiliations:** 1Division of Respiratory Medicine, Department of Pediatrics, University of California San Diego, La Jolla, CA 92093, USA; tstobdan@ucsd.edu (T.S.); njwen@ucsd.edu (N.J.W.); ylubo@health.ucsd.edu (Y.L.-B.); d2zhou@health.ucsd.edu (D.Z.); 2Department of Neurosciences, University of California San Diego, La Jolla, CA 92093, USA; 3Rady Children’s Hospital, San Diego, CA 92123, USA

**Keywords:** hypoxia, pupae, oxidative stress, epigenetic

## Abstract

Hypoxia not only plays a critical role in multiple disease conditions; it also influences the growth and development of cells, tissues and organs. To identify novel hypoxia-related mechanisms involved in cell and tissue growth, studying a precise hypoxia-sensitive time window can be an effective approach. *Drosophila melanogaster* has been a useful model organism for studying a variety of conditions, and we focused in this study on the life cycle stages of *Drosophila* to investigate their hypoxia sensitivity. When normoxia-grown flies were treated with 4% O_2_ at the pupa stage for 3, 2 and 1 day/s, the eclosion rates were 6.1%, 66.7% and 96.4%, respectively, and, when 4% O_2_ was kept for the whole pupa stage, this regimen was lethal. Surprisingly, when our hypoxia-adapted flies who normally live in 4% O_2_ were treated with 4% O_2_ at the pupa stage, no fly eclosed. Within the pupa stage, the pupae at 2 and 3 days after pupae formation (APF), when treated for 2 days, demonstrated 12.5 ± 8.5% and 23.6 ± 1.6% eclosion, respectively, but this was completely lethal when treated for 3 days. We conclude that pupae, at 2 days APF and for a duration of a minimum of 2 days, were the most sensitive to hypoxia. Our data from our hypoxia-adapted flies clearly indicate that epigenetic factors play a critical role in pupa-stage hypoxia sensitivity.

## 1. Introduction

Hypoxia plays a detrimental role in multiple disease conditions. A failure in adequately maintaining oxygen (O_2_) homeostasis leads to various pathological conditions, and some of them, e.g., cardiac ischemia, stroke, etc., are a leading cause of death in humans [1,2]. Although cells and tissues frequently encounter hypoxia, a series of molecular responses can maintain O_2_ homeostasis, for example, through the stabilization of hypoxia-inducible factors (HIFs), well-known master regulators of O_2_ homeostasis [3,4].

*Drosophila melanogaster* has been an important model organism in studying a number of conditions, including to identify hypoxia-related markers. Indeed, newer technologies have further helped us identify numerous hypoxia targets in embryos, larvae and adult flies [5,6,7]. However, a systematic study to identify the critical O_2_ level and critical sensitive stage that limit its normal development has never been attempted. Previous reports in embryos, larvae and adults have indicated significant variability in O_2_ sensitivity. For example, at 2% O_2_, there are reports of cell cycle arrest in embryos [8,9,10] and behavioral changes in larvae [11]. Similarly, we and others have previously reported that adult flies are extremely resistant to anoxia [12,13]. At the same time, none of the flies survived when kept in a 4% O_2_ environment for their entire life cycle, i.e., 0% pupa eclosion [6]. Given that the survival of the fittest is dictated by the most vulnerable stage of the life cycle, we believe identifying the critical hypoxia-sensitive phase will facilitate the identification of key genes and pathways involved in O_2_ sensing. This will also help us to enrich our understanding of the role that hypoxia plays during normal development and identify novel hypoxia-responding gene/s with therapeutic potential. In this study, we systematically examined different stages in the life cycle of *Drosophila melanogaster* to identify critical hypoxia-sensitive stages.

## 2. Results

### 2.1. Determination of the Critical O_2_ Level for Hypoxia Sensitivity

In order to determine an optimal O_2_ level that has a substantial but not lethal effect on flies, we exposed embryos from control flies, i.e., *w1118* and *y^1^v^1^*, and hypoxia-adapted (HA) flies to 3.5%, 3.7%, 4%, 5% and room-air O_2_ for 21 days. The 3.5% and 3.7% O_2_ environment were lethal for the pupae of control flies (>100 uneclosed pupae). As anticipated, the eclosion rate in HA flies was significantly higher, i.e., 53.9 ± 7.9% and 80.2 ± 3.9% at 3.5% and 3.7% O_2_, respectively (Figure 1 and Appendix A). At 4% O_2_, the eclosion rates were between 0.8% and 2% in the control flies and 76.9 ± 13% in HA flies, and at 5% O_2_, the eclosion rates were 11.1 ± 6% and 18.7 ± 1.9% in *w1118* and *y^1^v^1^* and 78.2 ± 8.2% in HA flies (Figure 1 and Appendix A). We used the 4% O_2_ level for subsequent screening.

### 2.2. Identification of the Critical Hypoxia-Sensitive Stage

To determine the critical hypoxia-sensitive stage, we treated flies with 4% O_2_ at different stages of their life cycle, i.e., embryo; first-, second- and third-instar larvae; and early, mid and late pupae. Figure 2 (pie chart in the top section) depicts the specific stage at which the flies were transferred to a 4% O_2_ chamber from normoxia (21% O_2_) or vice versa. Our results indicate that the pupa stage was the most critical hypoxia sensitivity stage. For example, there was no eclosion in the normoxia-grown flies when treated with 4% O_2_ at the pupa stage compared to >97% in hypoxia-grown (4% O_2_) control flies; i.e., control flies grown in 4% O_2_ from embryo to late third-instar larvae returned back to normoxia before the pupa stage (Figure 2 and Appendix A). Our results also confirm that 4% O_2_ does not have a significant impact on the embryos, as the number of pupae in the experimental groups where the embryos were pretreated with 4% O_2_ for up to 48 h and the normoxia groups, i.e., no hypoxia treatment (0 h), were similar (Appendix A). Similarly, the eclosion rates of these pupae were >90% (Appendix A). Interestingly, the rate of eclosion decreased as the starting point of 4% O_2_ treatment moved from 1 day after pupa formation (APF) to 3 days APF; i.e., the eclosion rate decreased from >99% when exposed to 4% O_2_ at 1 day APF to 66.1 ± 22.09% and 89 ± 7.04% at 2 days APF and 54.3 ± 11.33% and 66.2 ± 8.72% at 3 days AFP in *w1118* and *y^1^v^1^*, respectively (Figure 2 and Appendix A), indicating early pupae are less hypoxia-sensitive.

To see if the HA flies, which complete their entire life cycle in 4% O_2_, are protected from extreme hypoxia sensitivity of the pupae, we treated normoxia-grown hypoxia-adapted pupae at 4% O_2_. Surprisingly, similar to the *w1118* and *y^1^v^1^* controls, none of the HA pupae eclosed (Figure 3). This is in contrast to the 76.9 ± 13% eclosion observed when HA flies were kept in 4% O_2_ for their entire life cycle (Figure 1 and Appendix A). Presumably, epigenetic mechanisms govern hypoxia sensitivity during pupa development.

### 2.3. Critical Time Period for Pupa Hypoxia Exposure

To assess the duration of hypoxia exposure critical to detect a noticeable phenotype, we treated normoxia-grown pupae with 4% O_2_ for different time periods and at different stages, i.e., 1, 2 and 3 days APF. Although 24 h of exposure at 1, 2 and 3 days APF did not affect the eclosion rates (Appendix A), we observed a significant drop in the eclosion rate when the exposure duration was ≥2 days of hypoxia exposure (42.32 ± 9.2% at 2 days and 6.07 ± 1.8% at 3 days, Appendix A).

In order to determine the critical phase during the pupal period when flies are most sensitive to hypoxia, we treated HA pupae with 4% O_2_ for 2 and 3 days. For this, we used pupae aged 1 to 5 days APF (Section 4). There was no significant change in the eclosion rate of 1-day-APF pupae exposed to hypoxia for 3 days (Figure 4a and Appendix A). However, the eclosion rate dropped from 89.3 ± 5.4% to 0% (*p*-value < 0.001) when 2-day or 3-day-APF pupae were exposed to 4% O_2_ for 3 days. Similarly, the eclosion rate also dropped from 81.3 ± 12.7% to 12.5 ± 8.5% (*p*-value < 0.001) when 2-day-APF pupae were treated for 2 days in 4% O_2_ and to 23.6 ± 1.6% (*p*-value < 0.001) in 3-day-APF pupae treated with a similar hypoxia level (Figure 4b and Appendix A). Overall, this experiment clearly indicates that pupae at 2 to 3 days APF are in the phase most vulnerable to hypoxia stress in the entire life cycle of flies, and the minimum time duration is 2 days.

## 3. Discussion

A practical and effective method of identifying novel hypoxia-related markers is to first identify the critical hypoxia-sensitive phase, e.g., during development, and then to conduct focused experiments using the latest techniques to identify the molecular mechanisms. We and others used the embryo, larva or adult stages of *Drosophila melanogaster* to study various aspects of the hypoxia response, but systematic studies to identify the critical hypoxia-sensitive time window have rarely been conducted. The few studies with a somewhat similar objective have not pinpointed a specific time window, likely because of using a milder hypoxia (10% O_2_) treatment or due to considering a modest endpoint phenotype, i.e., body size, or the combination of both [14]. Here, we attempted to address this issue, and our hypoxia tolerance screening results have revealed early pupae as the most hypoxia-sensitive stage in the life cycle of *D. melanogaster*. Exposing flies to 4% O_2_ for 2 days during the early pupa stage, i.e., 2 days APF, led to ~69% drop in the eclosion rate, and a 3-day exposure period was lethal. Interestingly, even the HA flies, which normally live in 4% O_2_, were equally affected (Figure 3), which is interesting and unexpected. Surprisingly, we found that the effects of environmental stress during the pupal stage in insects have rarely been investigated as previous studies are largely focused on its impact on the embryo, larva or adult stages. This is despite the fact that metamorphosis involves a radical transition from a larval to an adult body form and function, making the pupa stage a highly stress-sensitive stage [15].

Previous studies have shown that hypoxia stress during the larval period negatively affects adult fitness, which includes a reduced lifespan [16,17], decreased body size [14,18] and reduced stress resistance [17] in adults. Although these studies primarily focused on the larval stage, some of the results support and strengthen our data, independently demonstrating a similar hypoxia-sensitive window [14,18]. For example, Heinrich et al. noted that brief hypoxia exposure during the late larval and mid-pupal periods (around 2 days APF) have the greatest effects on adult size [14], which is consistent with our current study on lethality, due to short hypoxia exposure during the same pupa period. Likewise, the pupal stage was found to be a critical stage beyond which the detrimental effects of hypoxia could not be reversed [18]. Besides hypoxia, among the limited studies that have discovered pupa sensitivity to other types of stress is the oriental fruit fly’s (*Bactrocera dorsalis*) response to heat exposure during the pupal stage, which negatively influences longevity and heat tolerance in adults [19]. Similarly, *Plutella xylostella* adults (a global insect pest), when heat-treated as pupae, always lay lower numbers of eggs [20]. As anticipated, the differences were only significant when the heat exposure was >16 h [20]. Similarly, a 24 h period of exposure to 10% O_2_ during the early pupal period significantly reduces adult size [14], which correlates with our observation of a sudden drop in the eclosion rate when the hypoxia exposure was ≥2 days.

While previous studies have shown that even brief 10% O_2_ exposure can have a significant impact on body size, we used the eclosion rate as the endpoint phenotype and we observed that a 5% O_2_ level has a significant impact on the eclosion rate; hence, we used O_2_ levels < 5% in our experiments. For example, when the naive flies were treated with 5% O_2_ for their entire developmental period, the eclosion rate was between 0% and 28%, but at 4% O_2_, none survived [6]. Additionally, the HA flies that we generated in our laboratory through a laboratory selection experiment were adapted to 4% O_2_, i.e., completed their entire life cycle in a 4% O_2_ environment [6]. Additionally, based on these considerations, we believe that 4% O_2_ constituted a clear distinction between naive (non-adapted) flies and HA flies. Additionally, because our objective was to identify the critical hypoxia-sensitive stage, which involved hypoxia treatment for a relatively shorter period of time (compared to the entire life cycle), we also investigated eclosion rates at 3.5% and 3.7% O_2_ levels to have further resolution of that particular critical level. Surprisingly, even at 4% O_2_, a shorter treatment during the pupa stage was detrimental.

It is surprising that the treatment of pupae of HA flies with 4% O_2_ was lethal (similar to the control flies), which is unexpected as they had been living and reproducing in a 4% O_2_ environment. This clearly indicates that a short period of exposure to hypoxia during the early pupa stage could effectively eliminate the genetic adaptation to the same environment that took these flies > 290 generations to achieve during a period of >15 years of laboratory selection [6,21]. This is an important finding, and it would have been impossible to infer or to propose this idea using naive flies. It primarily reveals a pre-developed trajectory whereby the tolerance of subsequent stages is dependent on the conditions during the preceding stages, remarkably overriding the impact of its genetic background. Importantly, it also illustrates the “rigidity” in adaptation, whereby the resistance to a stressful condition may not be applicable in parts to a specific developmental stage, i.e., genetic adaptation per se does not guarantee survival when the treatment is applied in different developmental stages. A potential explanation for this is that the beneficial effect of early life stress exposure can epigenetically influence gene expression for later life stages. In other words, the genome of HA flies remains the same; however, when exposed to a hypoxic environment early in life, the differences in gene expression arising during development render them hypoxia-tolerant. Recent studies have shown that environmental factors can epigenetically influence gene expressions across developmental stages and generations [22]. For example, previous studies have shown that in *Drosophila*, early life exposure to environmental factors like low-dose oxidants can increase longevity [23]. In addition, there are intrinsic factors, like transient sulfotransferase overexpression during larval development, that significantly increase lifespan in adult flies [24]. Interestingly, in contrast to these results, there are reports that also suggest that early hypoxia treatment can lead to reduced stress resistance or lifespan in the adult [16,17]. Such contradictory results are difficult to explain since such responses are dependent on the level of hypoxia, its duration and the stage at which the stress was applied. Importantly, since this experiment suggests epigenetic factors play a crucial role in the cellular response to hypoxia [25,26,27], the study of the critical window that we discovered here is especially important because of the likely involvement of non-genetic factors in otherwise hypoxia-tolerant HA flies. In addition, our results suggest that the hypoxia sensitivity experienced during the critical time window predisposes them to conditions later in life. Therefore, to identify molecular predispositions, we will need to study the transcriptome and epigenome of stages prior to pupa formation, both with and without hypoxia treatment.

Finally, our current study has therapeutic value in identifying mechanisms underlying hypoxia-driven epigenetic regulation [28]. For example, hypoxic conditions in tumors contribute to cancer development and progression [29,30], and recent studies have shown that epigenetic changes due to a tumor’s microenvironment have profound impacts on its progression [29,31]. Exploring the precise molecular mechanisms will surely enrich our understanding of the hypoxia response and further disease therapy.

## 4. Materials and Methods

### 4.1. Fly Rearing and Collection

The *Drosophila* stocks *w1118* (Stock# 3605) and *y^1^v^1^* (Stock# 1509) were obtained from the Bloomington Stock Center. The HA flies were generated using long-term experimental selection over many generations, and the details of the methods used have previously been reported [6,21]. Briefly, we first generated a heterogeneous parental population by pooling 27 wild-type isogenic lines. When these flies were treated at 4%, 6% and 8% O_2_, we noticed that the percentages of embryos reaching the adult stage were 0%, <10% and >80%, respectively. Therefore, the hypoxia selection was initiated at 8% O_2_. Subsequently, the O_2_ concentration was gradually decreased by ∼1% every 3 to 5 generations, and by the 32nd generation, the flies were able to complete their entire life cycle in 4% O_2_. Since then, the flies have completed >290 generations in this environment. Under all conditions, the flies were maintained on standard cornmeal diet and were kept in ambient conditions of 25 °C with a 12–12 light–dark cycle. For the hypoxia and normoxia experiments, the ambient temperature and humidity were maintained at 22–24 °C and 30–50%, respectively.

### 4.2. Critical O_2_ Level Determination

For all the hypoxia treatments, we used O_2_ control glove boxes (Coy Lab Products, Grass Lake, MI, USA) with an oxygen controller at 0–100% +/− 0.1% resolution. We kept 2–3-day-old adult flies in room air for 24 h for egg laying. Subsequently, the adult flies were transferred to a new vial, and the original vials with eggs/embryos were transferred to a hypoxia chamber for 21 days. Another batch of vials with the eggs/embryos was kept in room air. The number of eclosed pupae was counted after 21 days.

### 4.3. Hypoxia-Sensitive Stage Screening

The hypoxia-sensitive stage screening assay was performed using *w1118* and *y^1^v^1^* fly lines and included two sets of vials. In the first set, the adult flies were kept in room air in new vials for 48 h for egg laying. After removing the adults, the vials were transferred to a 4% O_2_ chamber when the flies were at specific stages of their life cycle. The second set included new vials with eggs laid in the 4% O_2_ chamber, and then the vials were transferred to room air at specific developmental stages. Under both conditions, the eclosion rate was quantified at the end of 21 days. For the “all stage” hypoxia treatment, the vials were kept in 4% O_2_ the entire time, including during the initial egg laying stage (Figure 2 and Appendix A). In order to detect the detrimental effect of hypoxia on specific embryo stages, we also moved the hypoxia-grown embryos to room air 0, 5, 8, 12, 24, 36 and 48 h after the egg laying period. To see if 4% O_2_ had an impact on the pupation, i.e., the number of pupae formed from eggs, we calculated the percentage of pupae formed from eggs/embryo.

### 4.4. Pupa Timeline Screening

After 2 h of egg laying, the adult flies were removed and kept in room air. Pupae were marked on the vials within 4 h of pupa formation, and the subsequent time was counted as the time “after pupae formation” (APF). These pupae were then treated with 4% O_2_ at different days APF and for different time durations, in days.

### 4.5. Statistical Analysis

Each group contained 3 to 5 technical replicates, and each experiment was performed at least two times. The results are presented as means ± standard deviations. Data comparison between groups was analyzed using a simple *t*-test. All statistics were calculated using Microsoft Excel version 2016. Changes were considered statistically significant if *p* < 0.05.

## Figures and Tables

**Figure 1 ijms-25-00710-f001:**
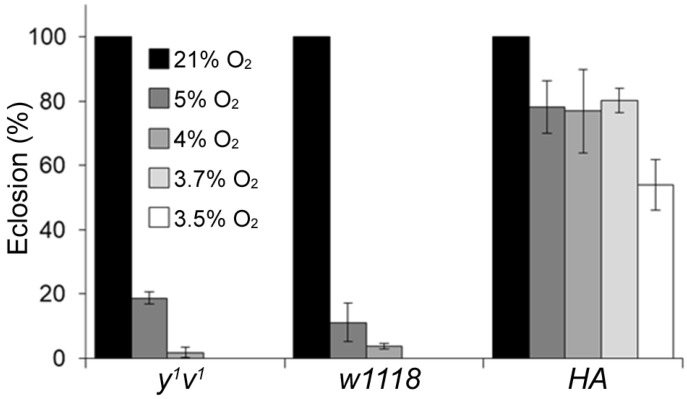
Eclosion rates of control and hypoxia-adapted (HA) flies when chronically exposed to different O_2_ levels. In the control flies (*w1118* and *y^1^v^1^* lines), no eclosion was detected in 3.5% and 3.7% O_2_ level environments, 0–5% in the 4% O_2_ environment and between 11% and 19% in the 5% O_2_ environment. In the HA flies, the eclosion rates were 53.9% at 3.5% O_2_ and >75% in 3.7%, 4% and 5% O_2_ environments.

**Figure 2 ijms-25-00710-f002:**
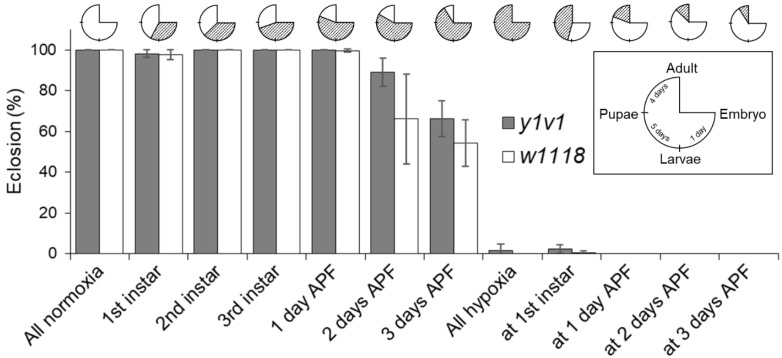
Screening for hypoxia-sensitive stage in the life cycle of *Drosophila melanogaster*. Hypoxia treated during 1st-, 2nd- or 3rd-instar larvae resulted in >97% eclosion. The eclosion rate dropped significantly from >99% at 1 day after pupa formation (APF) to 66.1% in *w1118* and 54.4% in *y^1^v^1^* at 3 days APF to no eclosion (0%) when normoxia-grown larvae or pupae were treated at 4% O_2_. (Top) Schematic of the experimental method, i.e., the life cycle stage (insert) at which the flies were treated with hypoxia. The shaded part indicates the time period during which flies were kept in 4% O_2_ and the unshaded part is the time spent in room air. Data presented in Appendix A.

**Figure 3 ijms-25-00710-f003:**
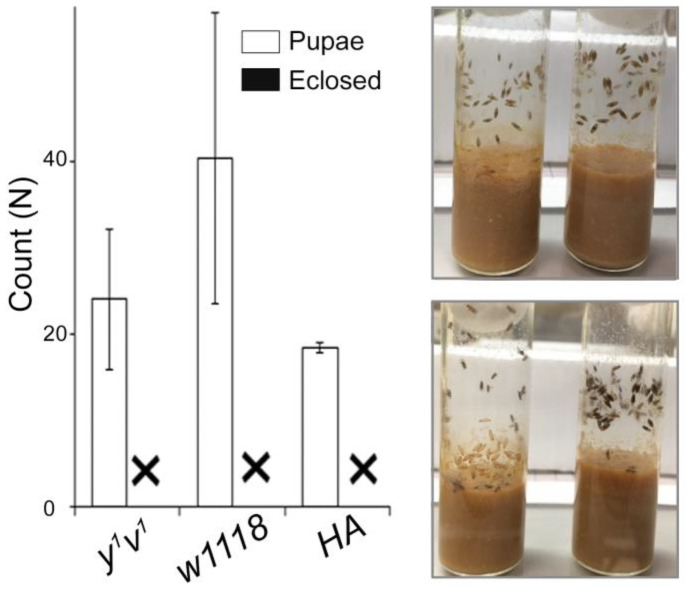
Eclosion rates of control and hypoxia-adapted flies when chronically exposed to 4% O_2_ APF. No eclosion (marked as ‘X’) was observed in the control fly lines (*w1118* and *y^1^v^1^*) or in the HA flies at 4% O_2_ (*p*-value < 0.001). Pictures show vials kept in 4% O_2_ throughout development (left) and only at the pupa stage (right) in *w1118* (top) and HA flies (bottom).

**Figure 4 ijms-25-00710-f004:**
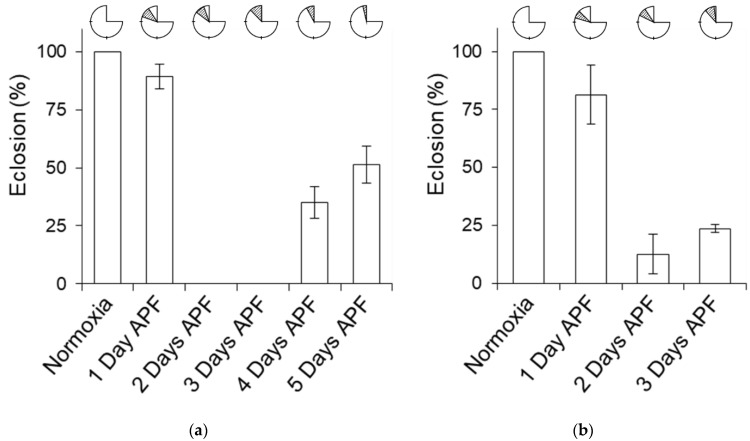
Hypoxia exposure of pupae for 2 days when they are 2 to 3 days APF is the critical hypoxia-sensitive phase. (**a**) The eclosion rates of normoxia-grown HA flies at 1 to 5 days APF kept for 3 days in 4% O_2_. No eclosion was seen in 2- and 3-day-APF pupae when treated with hypoxia for 3 days. Data presented in Appendix A. (**b**) Eclosion rates of normoxia-grown HA flies kept in hypoxia for 2 days. The age of pupae when treated with hypoxia were 1, 2 and 3 days APF. Data presented in Appendix A. The eclosion rates were <25% in 2- and 3-day-APF pupae. Top: the shaded part indicates the life cycle stage and the time period during which the flies were kept in 4% O_2_, and the unshaded part is the time spent in room air.

## Data Availability

Data are contained within the article or Appendix A.

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
