# Peer review of "The Pupa Stage Is the Most Sensitive to Hypoxia in Drosophila melanogaster"

_ijms, 2024, doi:10.3390/ijms25020710_

Round 1

Reviewer 1 Report

Comments and Suggestions for Authors

This intriguing study delves into the fascinating realm of hypoxia sensitivity during the life cycle stages of Drosophila melanogaster, shedding light on its impact on cell and tissue growth. The meticulous exploration of specific hypoxia-sensitive time-windows, particularly during the pupae stage, unravels compelling findings. The differential eclosion rates under varying oxygen concentrations offer valuable insights, with the most striking revelation being the heightened sensitivity of pupae at two days post-formation. The study's innovative approach, including the examination of hypoxia-adapted flies, adds depth to our understanding of epigenetic factors influencing pupae stage hypoxia sensitivity. Overall, this research contributes significantly to the field, emphasizing the importance of precise temporal considerations in unraveling the complex interplay between hypoxia and developmental processes in Drosophila. I have some concerns:

1. Figure1: It is best to have statistical results for this data.

2. What is the rationale behind relocating eggs/embryos to a hypoxic chamber for a duration of 21 days? What constitutes the ambient temperature during this process? In the event that the ambient temperature is a standard 25°C, how does the life cycle of fruit flies differ under low oxygen conditions? It is crucial to provide a thorough explanation of this phenomenon.

3. Statistical analysis: Was the data assessed for adherence to a normal distribution? If so, which method was employed for this evaluation?

Reviewer 2 Report

Comments and Suggestions for Authors

To authors

The manuscript presents a thorough systematic study to identify a critical level of O2, specifically the critical time window sensitive to hypoxia that limits normal larval development. This ceiling of O2 levels that limits hatching has revealed that early pupae are the most sensitive stage to hypoxia.

Questions:

1)     Lines 58 to 61: “At 4% O2 the eclosion rates were between 0.8% and 2% in the control flies and 76.9% in HA flies and at 5% O2 the eclosion  rates were 11.1% and 18.7% in the w1118 and y1v1, and 78.2% in HA flies (Figure 1). We used  4% O2 level for subsequent screening”.

To calculate the ceiling in oxygen levels the authors use a hypobaric hypoxia model (similar to altitude) below 21% of atmospheric O2. But have the authors analyzed the O2 saturation in the hemolymph? These data could provide clinical transfer values.

2)     Paragraph 2.1 (and the rest of the paragraphs with results): You show numerical data as % standard mean +/- standard deviation. In others you only show % data. Please, I would recommend to the authors an annex where a table of each figure is collected with the percentage data, mean +-/standard deviation, and p-values. You also do not indicate in the texts if there are significant differences.

3)     Lines 83 to 86: “Surprisingly, similar to the w1118 and y1v1 controls, none of the HA pupae eclosed (Figure 3). This is in contrast to the 76.9% eclosion when HA flies were kept in 4% O2 for its entire life cycle (Figure 1), presumably epigenetic mechanisms  governing hypoxia sensitivity during pupae development”.  

Lines 187 to 189: “This clearly indicates that a short-time exposure to hypoxia during early pupae could effectively eliminate the genetic adaptation to the same environment  that otherwise took these flies >290 generations during a period of >15 years of laboratory  selection (5, 6, 21)”.

So, in HA flies, both the pupation and hatching phases should have a high %. Figure 1 shows a high number of hatched ones and in figure 3 they do not hatch. What are the reasons for these changes between figures 1 and 3? The authors discuss epigenetic changes, please elaborate a little more than what is explained in the discussion.

In humans, during acclimatization the first thing that appears are changes in respiratory flow (increasing pulmonary ventilation) and increasing cardiac output. Later, the synthesis of the hormone erythropoietin appears, an increase in the number of erythrocytes and their entire hematocrit (including hemoglobin), together with angiogenesis and an increase in the number of mitochondria. I ask the authors, is it likely that moving the HA pupa in a state from normoxia to hypoxia may not activate the genetic mechanisms of mitochondrial overproduction in time?

4)     Could these results be transferred to human studies? Please make a small paragraph in the discussion about whether the animal model allows advances for future studies in humans.

Round 2

Reviewer 2 Report

Comments and Suggestions for Authors

I thank the authors for the quick response to clarify my questions and for providing new additional data. I also appreciate the reconsideration of my advice. 

Best regards.